# South Asian, Black and White ethnicity and the effect of potentially modifiable risk factors for dementia: A study in English electronic health records

**Naaheed Mukadam**[1]*, **Louise Marston**[2], **Gemma Lewis**[1], **Rohini Mathur**[3], **Ed Lowther**[4], **Greta Rait**[2], **Gill Livingston**[1]

**1** Division of Psychiatry, University College London, London, United Kingdom, **2** Primary Care & Population Health, University College London, London, United Kingdom, **3** Wolfson Institute of Population Health, Queen Mary University London, London, United Kingdom, **4** Advanced Research Computing Centre, University College London, London, United Kingdom

* n.mukadam@ucl.ac.uk

**Data Availability Statement:** Data cannot be shared publicly due to restrictions from CPRD. Others can access these data by applying to CPRD

## Abstract

### Introduction

We aimed to investigate ethnic differences in the associations of potentially modifiable risk factors with dementia.

### Methods

We used anonymised data from English electronic primary care records for adults aged 65 and older between 1997 and 2018. We used Cox regression to investigate main effects for each risk factor and interaction effects between each risk factor and ethnicity.

### Results

We included 865,674 people with 8,479,973 person years of follow up. Hypertension, dyslipidaemia, obesity and diabetes were more common in people from minority ethnic groups than White people. The impact of hypertension, obesity, diabetes, low HDL and sleep disorders on dementia risk was increased in South Asian people compared to White people. The impact of hypertension was greater in Black compared to White people.

### Discussion

Dementia prevention efforts should be targeted towards people from minority ethnic groups and tailored to risk factors of particular importance.

(https://cprd.com/data). The authors had no special privileges with regards to data access.

**Funding:** NM is funded by the Alzheimer's Society (AS-SF-18b-001). The funders had no role in study design, data collection and analysis, decision to publish, or preparation of the manuscript.

**Competing interests:** The authors have declared that no competing interests exist.

## Background

The number of people living with dementia worldwide is predicted to triple to over 150 million between 2019 and 2050 [1]. Age-specific dementia incidence and prevalence has declined in Western Europe and the United States but is rising in some lower and middle income countries [2]. Ageing populations mean there are more people with dementia in all countries [2]. The UK has a sizeable minority ethnic population (around 18%), of whom the largest is South Asian (Bangladeshi, Indian, Pakistani, Sri Lankan—6.9%) and the second largest group is Black (African or Afro-Caribbean– 4.0%). There has been growing interest in potentially modifiable risk factors for dementia as eliminating these could theoretically prevent around 40% of dementia [2]. Most risk factor studies, however, have been conducted in people of European origin and have not considered people from other ethnic groups [2]. In previous studies of cardiovascular disease, the association between fat deposition and diabetes [3] and between blood pressure and stroke [4] was found to be stronger in minority ethnic groups with some indication that combinations of risk factors may be particularly detrimental in these groups. The combination of hypertension and diabetes was found to be particularly detrimental for cardiovascular risk in both South Asian [4] and African Caribbean [5] people. Differences in arterial stiffness were reported to partly account for these variations in outcomes in people from minority ethnic groups [6, 7].

We previously investigated the interaction between risk factors and ethnicity with regards to dementia risk in the UK Biobank, a relatively small sample of volunteers who are healthier and more educated than the general UK population [8] and did not find evidence that ethnicity modified the association between risk factors and dementia risk. In this paper we report analyses using primary care records, a larger sample, representative of the general population. This addresses a need to investigate risk factors for dementia in UK minority ethnic populations and their impact on dementia risk compared to the White population to increase knowledge about the causes and prevention of dementia generally and in specific populations.

We aimed to investigate whether South Asian or Black ethnicity modifies the impact of established dementia risk factors on the risk of dementia in comparison to the White population.

## Methods

### Approvals

We used an anonymised dataset from the Clinical Practice Research Datalink (CPRD) which has National Research Ethics Service Committee (NRES) ethics approval for purely observational research using primary care data and established data linkages.

The study was approved by the Independent Scientific Advisory Committee (ISAC) of the Medicines and Healthcare Products Regulatory Agency (MHRA) (protocol 19_235).

We pre-registered this study's protocol prior to analyses (https://osf.io/jp6xu/).

### Datasets—Primary care

This study was carried out using the CALIBER © resource (https://www.ucl.ac.uk/health-informatics/caliber and https://www.caliberresearch.org/). CALIBER, led from the UCL Institute of Health Informatics, is a research resource providing validated electronic health record phenotyping algorithms and tools for national structured data sources [9, 10]. It uses anonymised electronic patient health records from CPRD, linked to Hospital Episode Statistics (HES) and Mortality Statistics in England. CPRD collects data from a UK-wide network of over 2,000 primary care practices and includes 50 million patients, of which 16 million are

currently registered active patients, with 25% of patients having ≥20 years of follow-up. CPRD patients are broadly representative of the general population [11]. Primary care physicians enter diagnoses (using Read codes) [12], medications and procedures in real time, as well as records of hospital correspondence. Read codes are a hierarchical recording system used to record clinical summary information. The codes are not limited to diagnostic and procedural codes, but also include symptoms, test results, screening and other areas. Disease phenotypes have been extensively validated in CALIBER [13]. CPRD provides two sets of data quality criteria: acceptability for patients and up to standard (UTS) time for practices which are used as a first step to selecting research-quality patients and periods of quality data recording. The acceptable patient metric is based on registration status, recording of events in the patient record, and valid age and sex. The UTS date is a practice-based quality metric based on the continuity of recording and the number of recorded deaths. The UTS date is calculated for each participating practice, corresponding to the latest date at which practices meet these minimum quality criteria [14].

## Hospital Episode Statistics (HES)

HES are electronic records of diagnoses and procedures from hospital inpatient, outpatient and emergency attendances in England. They are coded by trained non-clinical coders based on information from discharge summaries and record up to 20 diagnoses for each episode using the International Classification of Diseases version 10 (ICD10) coding system. We used data from all three HES datasets.

## Mortality statistics

Mortality statistics are derived from the national census of all deaths. Information is based on diagnoses recorded on death certificates and include the primary and up to 14 contributing causes of death.

## Study population

We included adults aged 65 and over with electronic health data between 1997 and 2018. As deprivation scores and linked electronic hospital records are country-specific and are only provided for practices in England which have consented to participate in the CPRD patient-level linkage scheme [15], we restricted analyses to practices from England which have consented to the patient-level linkage scheme (around 88% of practices [16]).

## Variables

### Dementia

We considered all-cause dementia as our main outcome. Dementia was defined as the record of one or more diagnostic codes in any of the three data sources (GP records, HES and mortality statistics) at any time and in any position (i.e., dementia in any of the recorded diagnoses in hospital admission or death record). We also classified people as having a diagnosis of dementia if they had at least one recorded prescription of rivastigmine, galantamine, donepezil or memantine, which are used to treat patients with Alzheimer's disease and sometimes patients with dementia in Parkinson's disease and dementia with Lewy bodies. Dementia in CPRD is validated [17], particularly all-cause dementia, with high positive predictive values (>75%) in studies examining all-cause dementia [18]. In HES, dementia has been validated against the gold standard of diagnosis of a specialist memory service [19]. Date of dementia was considered as the earliest date of any of the recorded medication or diagnostic codes.

## Ethnicity

Ethnicity is recorded in primary care data (CPRD), as well as HES. We used ethnicity as defined in either source of data at any time point. Where ethnicity is recorded in more than one of these databases, the agreement between ethnicities is high [20]. Where conflict existed between CPRD and HES, we prioritised the CPRD classification because CPRD ethnicity has been shown to reflect population percentages [20] whereas HES ethnicity is less accurate [21]. Ethnic categories reflected those recorded by the 2021 UK Census. The national census recognises 18 ethnic groups: White British, White Irish, White Gypsy/Roma, White Other, Asian Bangladeshi, Asian Indian, Asian Pakistani, Asian Chinese, Asian Other, Black African, Black Caribbean, Black other, Mixed White and Black Caribbean, Mixed White and Black African, Mixed White and Asian, Any other mixed background, Other (Arab), Other (any other ethnic group). Both CPRD and HES use these categories.

For this study we combined ethnicities, based on research showing clustering of low frequency genetic variants in people who originated in the same geographic location as follows [22]: White (all white groups); South Asian (Asian Bangladeshi, Indian, Pakistani); Black (Black British, Black African or Black Caribbean), Mixed (Any mixed background), Other (Chinese, Other Asian and any category classified as "other"). We included all ethnicities in the analyses to avoid bias but only reported results for White, South Asian, and Black participants.

## Dementia risk factors

We included as exposures all conditions that have consistently been shown to be associated with increased risk of dementia in previous studies [2, 23, 24] and were routinely recorded in primary care data. All risk factors have previously documented consistent associations with dementia with indications of a potential causal relationship [2]. We include hypertension, obesity, hearing loss, diabetes, smoking, depression, alcohol excess, dyslipidaemia (high total cholesterol or diagnostic codes for dyslipidaemia), low levels of High-Density Lipoprotein (HDL) cholesterol, high levels of Low-Density Lipoprotein (LDL) cholesterol, sleep disorders and traumatic brain injury as risk factors. Further details of how risk factors were defined and validated are in the S1 File. We counted exposure to a risk factor as occurring if the person met criteria for that risk factor prior to entry to the cohort.

## Confounders

Previous studies have adjusted for variables which might influence the association between risk factors and dementia including education, sex and other diseases. Many other studies adjust analyses for education but education is not measured in primary care data. We included baseline age, sex, and Index of Multiple Deprivation (IMD) [25] in all adjusted statistical models.

IMD is a composite measure based on postcode derived from indicators covering domains of material deprivation: income, employment, education and skills, health, housing, crime, access to services, and living environment. A higher score means more deprivation. We included IMD quintile in all models to adjust for deprivation.

Comorbidity–We did not adjust for other physical health conditions in our main analysis as many may be on the causal pathway between risk factors and dementia. For the sensitivity analysis exploring whether interactions could be due to comorbidity, we used a modified version of the Charlson index [26], removing diabetes and dementia from the index as these were respectively a risk factor and outcome. Calculation of this index is described in the S1 File.

## Statistical analysis

All analyses were conducted in Stata Version 17.0 except for the competing risks regression for which we used R, as detailed below. Participants entered the cohort on the latest of: the date they turned 65 years old, 1$^{st}$ January 1997, when the GP practice met data quality standards or one year after the date of registration with the GP practice to maximise [27] accuracy of recorded health conditions. Participants exited the cohort on the earliest of date of developing dementia, being lost to follow-up, date of death or end of data availability (31$^{st}$ December 2018).

We described baseline demographic and clinical characteristics at the start of the cohort for each participant across the whole sample and stratified by ethnicity (the three main ethnic groups of interest and missing ethnicity).

### Impact of risk factors in different ethnic groups

We conducted Cox regression models with time to dementia as the outcome. We first conducted Cox regression with each risk factor as exposure and adjusting for age. We then added sex and IMD quintile to each model, followed by ethnicity. Our final model included an interaction term between ethnicity and each risk factor. All models accounted for clustering, by GP practice using robust standard errors, as people attending the same practice are likely to receive similar care. If there was evidence of an interaction between the risk factor and ethnicity, we repeated the regression, stratifying by ethnicity, adjusting for age, sex and IMD quintile and including a term to account for clustering by GP practice.

### Missing data

As ethnicity was the principal variable of interest, our main analyses were complete case where data on outcome, ethnicity and confounders were all present. In a sensitivity analysis, we also conducted multiple imputation with chained equations to impute missing ethnicity data as those have been shown to be missing at random in similar analyses [28]. We analysed patterns of missingness and conducted regression analyses to explore associations between missing values of ethnicity and IMD and other variables in the dataset and concluded that the missing at random assumption was supported. We used the multiple imputation by chained equations algorithm [29] for multiple imputation of missing data in ethnicity and IMD quintile. Multiple imputation was performed in Stata version 17.0 using the "mi impute chained" command. For each incomplete variable, we constructed an imputation model conditional on variables in the main analysis (indicator of dementia, age, sex, all risk factors), the other incomplete variables (ethnicity, IMD) and other disease indicators recorded at any time in the patient records to improve the imputation model (myocardial infarction, stroke, chronic kidney disease). We created twenty imputed datasets and conducted sensitivity analyses for each of the regression analyses using multiply imputed data. Each imputed dataset was analysed identically, and results combined using Rubin's rules [30].

### Secondary analyses

We planned several analyses in addition to multiple imputation, prior to commencing analyses. The first was to repeat analyses but including the modified Charlson index to see if accounting for comorbidity altered the findings. The second was to conduct a competing risks regression, to account for death as a possible competing risk to dementia. Due to increased computational demands, this was not possible to run in Stata, so for this we used fastcmprsk v1.1.1, and mitools v2.4 (for the competing risks regression using imputed data) running on R version 4.2.1.

## Results

### Baseline characteristics

There were 5,056,123 adults with electronic health records eligible for inclusion. We excluded those with a diagnosis of dementia prior to the start of their records (N = 45,252) and those who were under the age of 65 at the start of data availability (N = 3,821,781). This left 1,189,090 participants (830,541 White, 13,082 South Asian, 9,166 Black, 13,860 people of other ethnicities and 322,441 with no recorded ethnicity). In those with a record of ethnicity, less than 1% had missing IMD. Complete case analysis included 865,674 people with 8,479,973 person years of follow up (Fig 1 shows the selection of participants for the study). Overall, 149,228 (12.6%) of the whole sample, 133,094 (16.0%) of White people, 1,128 (8.6%) of South Asian people, 1,107 (12.1%) of Black people and 1,347 (9.7%) of those from other ethnic groups developed dementia.

### Impact of risk factors

All risk factors except hypertension were associated with an increased risk of dementia. These results remained in the fully adjusted models (See Table 2). In fully adjusted models, hazard ratios (HRs) ranged from 1.15 for excess alcohol (95% CI 1.12–1.19), 1.20 for obesity (95% CI 1.16–1.25) and hearing loss (95% CI 1.17–1.23) to 2.26 for depression (95% CI 2.17–2.36). In fully adjusted models with interaction terms between risk factors and ethnicity, interaction terms were statistically significant for hypertension (HR 1.57, $p<0.0001$), obesity (HR 1.19, $p = 0.04$), diabetes (HR 1.22, $p = 0.001$), low HDL (HR 1.21, $p = 0.049$) and sleep disorders (HR 1.18, $p = 0.002$) in those of South Asian ethnicity compared with White people. Black ethnicity was associated with an increased risk of dementia in those with hypertension (HR 1.18, $p = 0.029$) and decreased risk of dementia in those with high LDL (HR 0.81, $p = 0.005$), compared with White people. For the interaction terms showing statistical significance, analyses stratified by ethnicity confirmed differences in risk factor impact across ethnic groups (see Table 3).

### Secondary analyses

We repeated all analyses using multiply imputed data. Main effects for all risk factors were similar to estimates using complete case data. In these analyses, interaction terms for all risk factors in South Asian and Black people were significantly greater than one, except excess alcohol in South Asians and hearing loss, depression and sleep disorders in Black people which did not reach statistical significance (Table 4).

To investigate whether worse physical health overall could account for associations, we repeated all analyses including the modified Charlson index in all analyses in addition to adjustment for age, sex and IMD. In these analyses, main effects for all risk factors were similar to regression analyses without the Charlson index. Interaction effects for hypertension, diabetes and sleep disorders remained statistically significant for South Asians. Interaction terms for Black people with hypertension and high LDL remained statistically significant (Table 5).

Our initial finding that hypertension was associated with a decreased risk of dementia was inconsistent with prior studies and with theoretical assumptions. We repeated analyses using lower age cut offs for hypertension diagnosis, as blood pressure tends to fall later in life and especially as part of the dementia prodrome. Analyses using hypertension diagnosed prior to age 60, 55 and 50 all gave the same results as hypertension diagnosed before age 65. We also conducted the same analyses only including people who had a recorded diagnosis of hypertension rather than including those with combination of diagnoses and antihypertensive

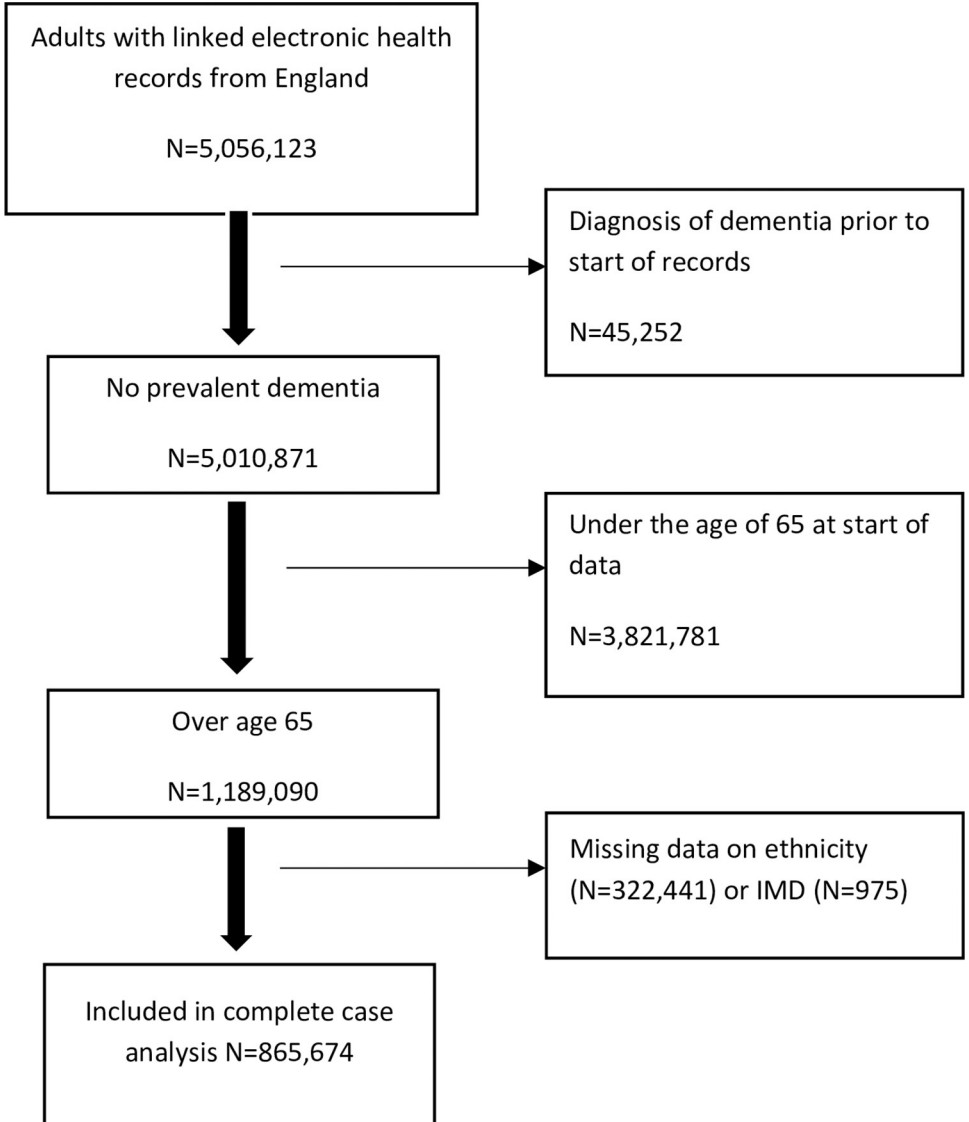

**Fig 1. Flow chart for included participants.** Baseline demographic characteristics and risk factor status for the sample are shown in Table 1. South Asian and Black people were on average younger than White people (mean age 69.4 and 69.7 respectively compared to 70.9) and there was a lower percentage of South Asian women (50.8%) compared to White (54.4%) and Black women (56.1%). A greater percentage of South Asian and Black people lived in more deprived areas (18.8% and 35.7% in the most deprived quintile) compared to White people (14.2% in most deprived quintile). Hypertension was most common in Black people (70.7%), followed by South Asians (66.0%) and with the lowest percentage of hypertension in White people (63.0%). The same pattern was seen for obesity– 34.2%, 22.8% and 19.2% in Black, South Asian and White people respectively. Hearing loss was least common in Black people with similar rates in White and South Asian people. Smoking rates were similar in White and South Asian people, but Black people had very low rates of smoking. Excess alcohol use was least common in South Asian people, and most common in White people. Diabetes was three times more common in South Asian people and more than twice as common in Black people than White people. Dyslipidaemia, low HDL and high LDL were all more common in South Asian and Black people than in White people. Sleep disorders were similarly prevalent in White and South Asian people and least common in Black people. Brain injury was recorded with similar frequency across all ethnic groups.

medications to see if potentially untreated hypertension would give different results. The results were similar to the original analyses (HR 0.63, 95% CI 0.62–0.63). We then performed competing risk analyses, including death as a competing risk and found hypertension was associated with an increased risk of dementia (HR 1.07, 95% CI 1.05–1.08). All other risk

**Table 1. Baseline demographic characteristics and risk factors of total complete case sample and divided into Black, South Asian and White ethnicity.**

| Characteristic | | All | White | South Asian | Black | Other | Missing ethnicity |
|---|---|---|---|---|---|---|---|
| | | N = 1,189,090 | N = 830,541 | N = 13,082 | N = 9,166 | N = 13,860 | N = 322,441 |
| **Age—mean(SD)** | | 70.9 (7.7) | 70.9 (7.5) | 69.4 (6.1) | 69.7 (6.1) | 70.1 (6.9) | 70.9 (8.3) |
| **N(%) female** | | 650,878 (54.7) | 452111 (54.4) | 6,648 (50.8) | 5,138 (56.1) | 7,556 (54.5) | 179,425 (55.7) |
| **IMD quintile** | 1 | 277776 (23.4) | 188859 (22.7) | 2,244 (17.2) | 610 (6.7) | 2,667 (19.2) | 83,396 (25.9) |
| | 2 | 275186 (23.1) | 193,998 (23.4) | 2,705 (20.7) | 921 (10.1) | 2,637 (19.0) | 74,925 (23.2) |
| | 3 | 256342 (21.6) | 180,684 (21.8) | 2,965 (22.7) | 1,686 (18.4) | 3,182 (23.0) | 67,825 (21.0) |
| | 4 | 208199 17.5) | 147,874 (17.8) | 2,686 (20.5) | 2,641 (28.8) | 3,066 (22.1) | 51,932 (16.1) |
| | 5* | 164973 (13.9) | 118,245 (14.2) | 2,459 (18.8) | 3,269 (35.7) | 2,276 (16.4) | 38,724 (12.0) |
| | Missing | 6614 (0.6) | 881 (0.1) | 23 (0.2) | 39 (0.4) | 32 (0.23) | 5,639 (1.8) |
| **Midlife hypertension** | | 623211 (52.4) | 523,376 (63.0) | 8,632 (66.0) | 6,482 (70.7) | 7,989 (57.6) | 76,732 (23.8) |
| **Midlife obesity** | | 203,875 (17.2) | 159,285 (19.2) | 2,979 (22.8) | 3,136 (34.2) | 2,814 (20.3) | 35,661 (11.1) |
| **Hearing loss** | | 133,258 (11.2) | 100,474 (12.1) | 1,588 (12.1) | 643 (7.0) | 1,359 (9.8) | 29,194 (9.1) |
| **Smoker** | | 318,124 (26.8) | 233,916 (28.2) | 3,627 (27.7) | 2,540 (7.7) | 4,024 (29.0) | 74,017 (23.0) |
| **Excess alcohol** | | 100,886 (8.5) | 76,714 (9.2) | 470 (3.6) | 425 (4.6) | 588 (4.2) | 22,689 (7.0) |
| **Diabetes** | | 113,912 (9.6) | 87,511 (10.5) | 4,855 (37.1) | 2,733 (29.8) | 3,184 (23.0) | 15,629 (4.9) |
| **Dyslipidaemia** | | 455,776 (38.3) | 340,811 (41.0) | 7,432 (56.8) | 4,951 (54.0) | 7,256 (52.4) | 95,326 (29.6) |
| **Low HDL** | | 70,399 (5.9) | 55,215 (6.7) | 2,320 (17.7) | 759 (8.3) | 1,659 (12.0) | 10,446 (3.2) |
| **High LDL** | | 240,150 (20.2) | 171,037 (20.6) | 4,565 (34.9) | 3,282 (35.8) | 4,606 (33.2) | 56,659 (17.6) |
| **Depression** | | 91,633 (7.7) | 69,392 (8.4) | 1,170 (8.9) | 623 (6.8) | 1,174 (8.5) | 19,274 (6.0) |
| **Sleep disorder** | | 290,297 (24.4) | 218,930 (26.4) | 3,328 (25.4) | 1,733 (18.9) | 3,110 (22.4) | 63,196 (19.6) |
| **Brain injury** | | 31,916 (2.7) | 24,384 (2.9) | 324 (2.5) | 194 (2.1) | 303 (2.2) | 6,711 (2.1) |

IMD = Index of Multiple Deprivation

* = most deprived

factors remained associated with an increased risk of dementia in these analyses. Interaction terms were not statistically significant for any risk factors in Black people but were greater than one and statistically significant for hearing loss, diabetes and low HDL in South Asian people (Table 6). Using competing risks regression in multiply imputed data gave similar main effects for all risk factors but in these analyses, most risk factors had statistically significantly interaction terms in both South Asian and Black people (S1 Table in S1 File). Competing risks regression in complete case analyses but including the Charlson index as an additional covariate also gave similar results (S2 Table in S1 File).

## Discussion

To our knowledge this is the first report of the impact of risk factors for dementia across Black, South Asian and White ethnic groups in a large sample representative of the general population. We find that potentially modifiable risk factors hypertension, obesity, diabetes, low HDL and sleep disorders confer a higher risk of dementia in South Asian, and hypertension in Black ethnicity groups compared to White people after adjusting for age, sex and deprivation. These associations remained after adjusting for comorbidity.

Our results are similar to previous findings in cardiovascular disease and stroke showing larger associations between ethnic group and risk factors in South Asian and African Caribbean people compared to people of European origin in the UK [31].

Our findings indicate that the same risk factors confer a higher risk of dementia in Black and South Asian origin people compared to White people, particularly for cardiovascular risk

**Table 2. Main effects and interaction effects.**

| Risk factors | Model 1 | | Model 2 | | Model 3 | | Fully adjusted* - main effects | | Interaction terms¥ | | | |
|---|---|---|---|---|---|---|---|---|---|---|---|---|
| | HR | 95% CI | HR | 95% CI | HR | 95% CI | HR | 95% CI | South Asian | | Black | |
| **Hypertension** | 0.89 | 0.87–0.91 | 0.88 | 0.86–0.90 | 0.86 | 0.84–0.88 | 0.68 | 0.66–0.69 | 1.57 | <0.0001 | 1.18 | 0.029 |
| **Obesity** | 1.32 | 1.27–1.37 | 1.31 | 1.26–1.36 | 1.30 | 1.25–1.35 | 1.20 | 1.16–1.25 | 1.19 | 0.04 | 0.92 | 0.287 |
| **Hearing loss** | 1.25 | 1.22–1.29 | 1.27 | 1.23–1.30 | 1.25 | 1.22–1.29 | 1.20 | 1.17–1.23 | 1.17 | 0.076 | 0.91 | 0.398 |
| **Smoker** | 1.40 | 1.35–1.46 | 1.41 | 1.36–1.47 | 1.40 | 1.35–1.46 | 1.35 | 1.30–1.41 | 1.00 | 0.975 | 0.96 | 0.639 |
| **Excess alcohol** | 1.12 | 1.09–1.16 | 1.19 | 1.16–1.23 | 1.19 | 1.15–1.23 | 1.15 | 1.12–1.19 | 0.97 | 0.887 | 1.12 | 0.503 |
| **Diabetes** | 1.66 | 1.61–1.71 | 1.68 | 1.63–1.73 | 1.66 | 1.61–1.71 | 1.52 | 1.48–1.57 | 1.22 | 0.001 | 1.06 | 0.496 |
| **Dyslipidaemia** | 1.46 | 1.41–1.51 | 1.46 | 1.41–1.51 | 1.46 | 1.41–1.51 | 1.36 | 1.31–1.40 | 1.09 | 0.24 | 0.94 | 0.399 |
| **Low HDL** | 1.59 | 1.49–1.71 | 1.66 | 1.55–1.78 | 1.65 | 1.54–1.77 | 1.52 | 1.42–1.62 | 1.21 | 0.049 | 1.04 | 0.771 |
| **High LDL** | 1.39 | 1.32–1.46 | 1.39 | 1.32–1.46 | 1.39 | 1.32–1.46 | 1.33 | 1.26–1.40 | 0.98 | 0.823 | 0.81 | 0.005 |
| **Depression** | 2.44 | 2.34–2.55 | 2.41 | 2.31–2.51 | 2.39 | 2.29–2.50 | 2.26 | 2.17–2.36 | 1.08 | 0.497 | 0.89 | 0.402 |
| **Sleep disorder** | 1.25 | 1.23–1.28 | 1.24 | 1.21–1.27 | 1.23 | 1.20–1.25 | 1.15 | 1.13–1.18 | 1.18 | 0.002 | 0.94 | 0.393 |
| **Brain injury** | 1.60 | 1.54–1.67 | 1.61 | 1.55–1.68 | 1.60 | 1.53–1.66 | 1.56 | 1.50–1.63 | 1.28 | 0.185 | 1.07 | 0.717 |

Model 1 –adjusted for age. Model 2 –adjusted for age and sex. Model 3 –adjusted for age, sex and Index of Multiple Deprivation (IMD).

\* = adjusted for age, sex, IMD and ethnicity

¥ = interaction term between ethnicity and risk factor, adjusted for age, sex and IMD. 95% CI = 95% confidence interval

factors such as lipids, hypertension, diabetes and obesity. This was particularly the case for the South Asian group. This may account for previous findings of a younger age of dementia diagnosis in minority ethnic groups, indicating greater susceptibility [32] to dementia and could be related to shorter survival after dementia as found in this cohort previously [33]. Our additional analyses indicated it is not likely to be because people from minority ethnic groups with these risk factors are in generally poorer health. It may be that risk factor severity is greater in minority ethnic groups who have a risk factor (e.g., higher average blood pressure in those with hypertension) or that risk factors have been present for longer. It may also be that factors such as delays in healthcare provision or suboptimal treatment due to discrimination account for or contribute to the difference in impact [34]. These interaction effects were not seen in our previous analyses [8] which had a relative lack of power with smaller numbers of people from minority ethnic groups who developed dementia. Given the representativeness of primary care patients, findings in this paper are more likely to be generalisable to the general population. This means that not only are some risk factors more common in minority ethnic groups, but their impact is increased compared to White people indicating a doubly increased risk.

**Table 3. Stratified analysis for risk factors with statistically significant interaction terms.**

| Risk factors | White | | | South Asian | | | Black | | |
|---|---|---|---|---|---|---|---|---|---|
| | HR | LCL | UCL | HR | LCL | UCL | HR | LCL | UCL |
| **Hypertension** | 0.67 | 0.66 | 0.68 | 1.04 | 0.91 | 1.19 | 0.81 | 0.71 | 0.93 |
| **Obesity** | 1.20 | 1.18 | 1.23 | 1.43 | 1.23 | 1.67 | 1.11 | 0.96 | 1.28 |
| **Diabetes** | 1.52 | 1.49 | 1.55 | 1.86 | 1.65 | 2.10 | 1.59 | 1.41 | 1.80 |
| **Low HDL** | 1.51 | 1.46 | 1.56 | 1.80 | 1.52 | 2.14 | 1.47 | 1.13 | 1.91 |
| **High LDL** | 1.34 | 1.31 | 1.36 | 1.29 | 1.12 | 1.49 | 1.05 | 0.91 | 1.22 |
| **Sleep disorder** | 1.15 | 1.14 | 1.17 | 1.37 | 1.22 | 1.55 | 1.09 | 0.95 | 1.26 |

All adjusted for age, sex and Index of Multiple Deprivation (IMD). LCL = lower 95% confidence limit, UCL = upper 95% confidence limit

**Table 4. Regression analyses using multiply imputed data, adjusted for age, sex and Index of Multiple Deprivation (IMD).**

| Risk factors | Main effects | | | Interaction terms | | | |
| --- | --- | --- | --- | --- | --- | --- | --- |
| | | | | South Asian | | Black | |
| | HR | LCL | UCL | HR | p value | HR | p value |
| **Hypertension** | 0.85 | 0.84 | 0.86 | 3.03 | <0.0001 | 3.10 | <0.0001 |
| **Obesity** | 1.28 | 1.23 | 1.33 | 1.59 | <0.0001 | 1.45 | <0.0001 |
| **Hearing loss** | 1.26 | 1.23 | 1.29 | 1.41 | 0.001 | 1.24 | 0.09 |
| **Smoker** | 1.39 | 1.34 | 1.45 | 1.32 | <0.0001 | 1.47 | <0.0001 |
| **Excess alcohol** | 1.18 | 1.14 | 1.22 | 1.27 | 0.17 | 1.66 | 0.002 |
| **Diabetes** | 1.65 | 1.60 | 1.70 | 1.68 | <0.0001 | 1.74 | <0.0001 |
| **Dyslipidaemia** | 1.45 | 1.40 | 1.50 | 1.53 | <0.0001 | 1.59 | <0.0001 |
| **Low HDL** | 1.63 | 1.51 | 1.74 | 1.71 | <0.0001 | 1.53 | 0.001 |
| **High LDL** | 1.38 | 1.31 | 1.45 | 1.36 | <0.0001 | 1.34 | <0.0001 |
| **Depression** | 2.38 | 2.28 | 2.48 | 1.43 | 0.001 | 1.31 | 0.06 |
| **Sleep disorder** | 1.23 | 1.20 | 1.25 | 1.39 | <0.0001 | 1.12 | 0.19 |
| **Brain injury** | 1.58 | 1.52 | 1.65 | 1.64 | 0.01 | 1.53 | 0.041 |

LCL = lower 95% confidence limit, UCL = upper 95% confidence limit

If risk factors increase dementia risk more in minority ethnic groups than in White people, this deserves further study to elucidate possible mechanisms. Sometimes this may be because of severity. There also should be greater emphasis on risk factor measurement and treatment to improve dementia prevention in these groups. In 2009, the NHS Health Check 5-yearly programme started in England, inviting those aged 40 and over to have their health assessed with the aim of identifying and addressing cardiovascular risk factors [35]. These could be a useful tool in addressing dementia risk but so far uptake has been poor [36], ethnicity is often not recorded [37] and patients have reported lack of clarity over risk scores and little support with behaviour change necessary to address their risk factors [38]. More work needs to be done to improve uptake, achieve more personalised risk targeting and have health impact. Addressing

**Table 5. Regression analyses including adjustment for modified Charlson index, as well as age, sex and Index of Multiple Deprivation (IMD).**

| Risk factor | Fully adjusted—main effects | | | Interaction terms | | | |
| --- | --- | --- | --- | --- | --- | --- | --- |
| | | | | South Asian | | Black | |
| | HR | LCL | UCL | HR | p value | HR | p value |
| **Hypertension** | 0.72 | 0.71 | 0.74 | 1.57 | <0.0001 | 1.20 | 0.02 |
| **Obesity** | 1.25 | 1.20 | 1.30 | 1.18 | 0.06 | 0.89 | 0.15 |
| **Hearing loss** | 1.21 | 1.18 | 1.24 | 1.17 | 0.08 | 0.94 | 0.59 |
| **Smoker** | 1.38 | 1.33 | 1.44 | 1.02 | 0.83 | 0.97 | 0.68 |
| **Excess alcohol** | 1.15 | 1.12 | 1.19 | 1.00 | 0.98 | 1.10 | 0.56 |
| **Diabetes** | 1.60 | 1.56 | 1.65 | 1.22 | 0.001 | 1.05 | 0.56 |
| **Dyslipidaemia** | 1.41 | 1.36 | 1.46 | 1.09 | 0.24 | 0.93 | 0.27 |
| **Low HDL** | 1.57 | 1.47 | 1.69 | 1.20 | 0.06 | 1.04 | 0.79 |
| **High LDL** | 1.37 | 1.30 | 1.44 | 0.96 | 0.64 | 0.79 | 0.002 |
| **Depression** | 2.29 | 2.20 | 2.39 | 1.08 | 0.50 | 0.92 | 0.56 |
| **Sleep disorder** | 1.17 | 1.15 | 1.20 | 1.21 | 0.001 | 0.95 | 0.44 |
| **Brain injury** | 1.56 | 1.50 | 1.62 | 1.31 | 0.13 | 1.12 | 0.55 |

LCL = lower 95% confidence limit, UCL = upper 95% confidence limit

**Table 6. Competing risks analysis, adjusted for age, sex and Index of Multiple Deprivation (IMD).**

| Risk factors | Main effects | | | Interaction terms | | | |
| --- | --- | --- | --- | --- | --- | --- | --- |
| | | | | South Asian | | Black | |
| | HR | LCL | UCL | HR | p value | HR | p value |
| **Hypertension** | 1.07 | 1.05 | 1.08 | 1.19 | 0.11 | 0.82 | 0.08 |
| **Obesity** | 1.32 | 1.30 | 1.34 | 1.14 | 0.28 | 0.88 | 0.25 |
| **Hearing loss** | 1.21 | 1.19 | 1.24 | 1.28 | 0.02 | 1.14 | 0.43 |
| **Smoker** | 1.11 | 1.09 | 1.13 | 1.09 | 0.43 | 1.00 | 0.99 |
| **Excess alcohol** | 1.09 | 1.06 | 1.12 | 0.94 | 0.84 | 0.58 | 0.08 |
| **Diabetes** | 1.16 | 1.14 | 1.19 | 1.51 | <0.0001 | 1.15 | 0.17 |
| **Dyslipidaemia** | 1.50 | 1.48 | 1.52 | 0.92 | 0.42 | 0.91 | 0.34 |
| **Low HDL** | 1.32 | 1.28 | 1.36 | 1.32 | 0.03 | 0.90 | 0.55 |
| **High LDL** | 1.52 | 1.50 | 1.55 | 0.85 | 0.14 | 0.85 | 0.11 |
| **Depression** | 1.58 | 1.55 | 1.62 | 1.06 | 0.69 | 0.94 | 0.72 |
| **Sleep disorder** | 1.03 | 1.01 | 1.04 | 1.23 | 0.06 | 0.87 | 0.26 |
| **Brain injury** | 1.44 | 1.38 | 1.49 | 0.93 | 0.83 | 0.74 | 0.36 |

LCL = lower 95% confidence limit, UCL = upper 95% confidence limit

discrimination within healthcare systems and ensuring treatment for risk factors is tailored according to the individual are essential to correct current imbalances and provide more equitable care.

Strengths of this study include a large sample size of nearly one million individuals, which is broadly representative of the general population and validated measures of risk factors and dementia. In addition, people were able to be followed until death. Our competing risk analysis showing that hypertension is a risk and initially appeared protective because of people dying before they developed dementia is important and may explain some contradictory findings about hypertension previously reported [39]. We found higher levels of many cardiovascular risk factors in South Asian and Black people compared to White people, particularly diabetes which was more common in South Asian people than in any other group and similar to findings in a population based cohort study, suggesting recording levels are broadly unbiased [40]. A limitation is the missing data on ethnicity, but analyses from multiply imputed data were similar to the complete case analysis, which strengthens the findings. Additionally, we used a postcode-based measure of deprivation (not based on individual level measures such as income) but it does take into account a complex mix of factors. We relied on routinely collected electronic health record data which may not measure or record some risk factors or diagnoses or not be up to date. Another limitation is classification of risks as either present or absent without considering their severity. However some risk factors are less common in the minority communities, for example excess alcohol [41] and cigarette smoking [42] and it seems unlikely that they are more severe. Others, like obesity, high LDL and hypertension may be more severe in minority ethnic populations which is more reason to target them. Depression consistently had large effect sizes but it is not possible to rule out that this could be due to reverse causality as it was counted as a risk factor regardless of when it occurred prior to dementia diagnosis. It is also not possible to rule out residual confounding in observational studies so it may be that this accounts for findings. Finally we did not have information on individual-level education, social isolation, air pollution and physical activity all of which are established dementia risk factors [2].

Overall, we found evidence that minority ethnic status confers a greater risk of dementia for many risk factors compared to White people. Hypertension, obesity, diabetes and low

HDL seemed particularly important in South Asians and hypertension relatively more important in Black people. Dementia prevention efforts should prioritise people from South Asian and Black groups, co-designing education, policy and behavioural interventions with these communities to ensure they are acceptable, feasible and scalable.

## Supporting information

**S1 Checklist. The RECORD statement–Checklist of items, extended from the STROBE statement, that should be reported in observational studies using routinely collected health data.**
(DOCX)

**S1 File. Supplementary tables and analyses.**
(DOCX)

## Author Contributions

**Conceptualization:** Naaheed Mukadam, Gemma Lewis, Rohini Mathur, Greta Rait, Gill Livingston.

**Data curation:** Naaheed Mukadam.

**Formal analysis:** Naaheed Mukadam, Louise Marston, Ed Lowther, Gill Livingston.

**Funding acquisition:** Naaheed Mukadam.

**Investigation:** Naaheed Mukadam.

**Methodology:** Gemma Lewis, Rohini Mathur, Greta Rait, Gill Livingston.

**Project administration:** Naaheed Mukadam.

**Supervision:** Gill Livingston.

**Writing – original draft:** Naaheed Mukadam.

**Writing – review & editing:** Naaheed Mukadam, Louise Marston, Gemma Lewis, Rohini Mathur, Ed Lowther, Greta Rait, Gill Livingston.

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
