## [Decision Letter · Decision Letter 0]

26 Jun 2023

PONE-D-23-12899South Asian, Black and White ethnicity and the effect of potentially modifiable risk factors for dementia: a study in English electronic health recordsPLOS ONE

Dear Dr. Mukadam,

Thank you for submitting your manuscript to PLOS ONE. After careful consideration, we feel that it has merit but does not fully meet PLOS ONE’s publication criteria as it currently stands. Therefore, we invite you to submit a revised version of the manuscript that addresses the points raised during the review process.

We look forward to receiving your revised manuscript.

Kind regards,

Taofiki Ajao Sunmonu

Academic Editor

PLOS ONE

Additional Editor Comments:

The article dealt with an important medical concerns on dementia among three racial groups in an English electronic health records and this study could help in individualizing medical management s of Dementia among these racial groups. The authors need to address the concerns of the eminent reviewers who reviewed this article to improve its quality. I wish tto congratulate the authors for a thorough and well done work

Reviewers' comments:

Reviewer's Responses to Questions

**Comments to the Author**

1. Is the manuscript technically sound, and do the data support the conclusions?

Reviewer #1: No

Reviewer #2: Yes

2. Has the statistical analysis been performed appropriately and rigorously? 

Reviewer #1: Yes

Reviewer #2: Yes

3. Have the authors made all data underlying the findings in their manuscript fully available?

Reviewer #1: No

Reviewer #2: Yes

4. Is the manuscript presented in an intelligible fashion and written in standard English?

Reviewer #1: Yes

Reviewer #2: Yes

5. Review Comments to the Author

Reviewer #1: Thank you for the opportunity to review this interesting paper. My comments are:

1. The methods section refer to study population aged 18 and above; this is at odds with the rest of the paper, including abstract where it is reported that only those 65 and above were included

2. In the dementia risk factors it is unclear if mid-life diagnosis relates to hypertension only or all risk factors; suggest re-write

3. The authors used IMD to account for deprivation but it is unclear how they accessed education levels which is part of the index; in the confounders section they mention they had no access to education levels

4. Treatment for the various modifiable risk factors is an important potential confounder but it does not appear to be part of the model at large. Establishing whether treatment (e.g. current or cumulative dose exposure to medication – see doi: 10.1002/trc2.12268 for an example of how this has been done for diabetes treatment) for these risk factors differs in its efficacy re dementia risk appears to me to be a valid research question. How do the authors justify this omission? This appears to only have been done for hypertension – it is unclear why this was not done for other conditions and why the cumulative dose was not taken into account.

5. The authors mention that literature indicates that dementia has an earlier onset and is more frequent in ethnic minorities. I would have expected that this is confirmed by the current analysis – it is unclear why this was not reported?

The introduction and discussion are appropriate for the subject area.

Reviewer #2: This is a well written manuscript highlighting differential impact of established risk factors for dementia by ethnicity status in a UK population. The authors should be congratulated for a thorough write up and acknowledgement of limitations to their work. Well done.

6. PLOS authors have the option to publish the peer review history of their article (what does this mean?). If published, this will include your full peer review and any attached files.

Reviewer #1: No

Reviewer #2: **Yes: **Professor Fred Stephen Sarfo

---

## [Author Response · Author response to Decision Letter 0]

24 Jul 2023

Reviewer #1: Thank you for the opportunity to review this interesting paper. My comments are:

1. The methods section refer to study population aged 18 and above; this is at odds with the rest of the paper, including abstract where it is reported that only those 65 and above were included

- Thank you for pointing this out. We have changed it to specify that we only included those aged 65 and over.

2. In the dementia risk factors it is unclear if mid-life diagnosis relates to hypertension only or all risk factors; suggest re-write

- We have now taken out “midlife” for hypertension and obesity as all risk factors were measured before the age of 65 and therefore all are technically midlife. We had only specified this as hypertension and obesity are specifically risk factors in midlife as both blood pressure and BMI decline in later life generally. However, as our cohort started at age 65, this distinction is irrelevant. 

3. The authors used IMD to account for deprivation but it is unclear how they accessed education levels which is part of the index; in the confounders section they mention they had no access to education levels

- IMD is provided by CPRD as a composite score which includes area level measures of education. We have described this in the methods section. Unfortunately individual level education is not recorded in primary care. We have now specified in the discussion that we did not have individual-level education to differentiate it from area-level education which is used to calculate IMD.

4. Treatment for the various modifiable risk factors is an important potential confounder but it does not appear to be part of the model at large. Establishing whether treatment (e.g. current or cumulative dose exposure to medication – see doi: 10.1002/trc2.12268 for an example of how this has been done for diabetes treatment) for these risk factors differs in its efficacy re dementia risk appears to me to be a valid research question. How do the authors justify this omission? This appears to only have been done for hypertension – it is unclear why this was not done for other conditions and why the cumulative dose was not taken into account.

- Unfortunately treatment for health conditions is part of the validated criteria to identify people with those conditions, including hypertension and diabetes so it was not possible to identify differences in treatment in this paper. This may be possible in future work where treatment can be looked at in a more detailed way.

5. The authors mention that literature indicates that dementia has an earlier onset and is more frequent in ethnic minorities. I would have expected that this is confirmed by the current analysis – it is unclear why this was not reported?

- We have previously used this cohort to report age of diagnosis and frequency of dementia in different ethnic groups. We have now added this reference to the manuscript.

The introduction and discussion are appropriate for the subject area.

- Thank you

Reviewer #2: This is a well written manuscript highlighting differential impact of established risk factors for dementia by ethnicity status in a UK population. The authors should be congratulated for a thorough write up and acknowledgement of limitations to their work. Well done.

- Thank you

---

## [Editor Report · Decision Letter 1]

28 Jul 2023

South Asian, Black and White ethnicity and the effect of potentially modifiable risk factors for dementia: a study in English electronic health records

PONE-D-23-12899R1

Dear Dr. Mukadam,

We’re pleased to inform you that your manuscript has been judged scientifically suitable for publication and will be formally accepted for publication once it meets all outstanding technical requirements.

Kind regards,

Taofiki Ajao Sunmonu

Academic Editor

PLOS ONE

Additional Editor Comments (optional):

Great works.
---

## [Editor Report · Acceptance letter]

15 Sep 2023

PONE-D-23-12899R1 

South Asian, Black and White ethnicity and the effect of potentially modifiable risk factors for dementia: a study in English electronic health records 

Dear Dr. Mukadam:

I'm pleased to inform you that your manuscript has been deemed suitable for publication in PLOS ONE. Congratulations! Your manuscript is now with our production department. 

Kind regards, 

on behalf of

Dr. Taofiki Ajao Sunmonu 

Academic Editor

PLOS ONE